# Taming contact line instability for pattern formation

A. Deblais[1], R. Harich[1], A. Colin[2] & H. Kellay[1]

Coating surfaces with different fluids is prone to instability producing inhomogeneous films and patterns. The contact line between the coating fluid and the surface to be coated is host to different instabilities, limiting the use of a variety of coating techniques. Here we take advantage of the instability of a receding contact line towards cusp and droplet formation to produce linear patterns of variable spacings. We stabilize the instability of the cusps towards droplet formation by using polymer solutions that inhibit this secondary instability and give rise to long slender cylindrical filaments. We vary the speed of deposition to change the spacing between these filaments. The combination of the two gives rise to linear patterns into which different colloidal particles can be embedded, long DNA molecules can be stretched and particles filtered by size. The technique is therefore suitable to prepare anisotropic structures with variable properties.

[1] LOMA, Laboratoire Ondes et Matiere d'Aquitaine (UMR 5798), Universite de Bordeaux—CNRS, 33405 Talence, France. [2] ESPCI, CNRS, SIMM UMR 7615, 11 rue Vauquelin, 75005 Paris, France. Correspondence and requests for materials should be addressed to H.K. (email: hamid.kellay@u-bordeaux.fr).

Coating surfaces with Newtonian fluids and complex fluids is essential in numerous industrial processes such as making functional or flexible thin films with applications spanning microfluidics, sensor arrays[1], electronics and biomedical devices[2]. Different coating techniques exist, such as dip and spin coating, rigid or flexible blade coating. The stability of the contact line between the fluid and the surface plays a key role in the control of the final deposit[3,4]. An example of an instability that occurs for advancing contact lines is the fingering instability, which takes place when a thin liquid film flows down an inclined plane. Previous work has shed light on this instability both experimentally[5,6] and theoretically[7,8], whereas others offer different interesting ways to control the fingering[9] and thus the coating patterns.

While much work has focused on understanding the instability of an advancing contact line driven by gravity[10,11] or temperature gradients[12], less work has focused on the case of receding contact lines, as in the case of dip or blade coating, which are supposedly stable for Newtonian liquids[13]. Despite this stability with respect to undulations and formation of fingers, receding contact lines can show the formation of particular patterns as they are prone to cusp formation producing triangular-like films with properties set by the receding velocity of the front. This has been demonstrated in dip-coating experiments[14], as well as in experiments using drops flowing down an inclined plane where the rear of the drop produces cusps[15]. These cusps give rise to the formation of rivulets breaking up into even smaller droplets[16]. The conditions for the formation of such structures have been studied in detail and are understood[15–18], at least partially.

Here, we show that the instability of a receding contact line towards cusp and droplet formation can be tamed through the use of viscoelastic fluids to produce linear patterns of variable spacings. These patterns can then be used to embed different colloidal particles, stretch long DNA molecules and filter particles by size. The preparation of tunable patterns of oriented well-spaced lines with different embedded objects can be useful in many applications.

## Results

**Filament formation.** Here, and by using a viscoelastic polymer solution, a non-Newtonian fluid, the receding contact line obtained in blade-coating experiments shows the formation of cusp-like patterns, resembling their Newtonian counterparts, but which give rise to long slender filaments instead of droplets. Figure 1 illustrates the patterns obtained with a high-molecular-weight polymer solution in coating experiments using a flexible blade (polyacrylamide (PAM), see Methods and Supplementary Figs 1 and 2). Long, slender and spatially organized filaments are obtained over large extended areas of the surface and for a wide range of velocities. The typical spacing between the filaments is an increasing function of the coating velocity, Fig. 1, in stark contrast with the fingering instability of advancing contact lines where the spacing between fingers is a decreasing function of this velocity.

**Phase diagram for filament formation.** To gain insight into this filament formation, we compare in Fig. 2 the observed behaviour with that of a Newtonian fluid, Glycerine. The contact angle of the polymer solution and that of Glycerine on the surface used is close to 90° (Supplementary Fig. 3 and Supplementary Table 1). It is expected that in such experiments the fluid coats the surface with a homogeneous film at velocities exceeding a certain threshold (Supplementary Note 1)[4,19]. Below this threshold, neither solution produces homogeneous films. In general, a capillary number, comparing viscous to capillary stresses, $Ca = \eta V / \sigma$ is defined, where $\eta$ is the viscosity of the solution, $V$ is the velocity of the substrate and $\sigma$ is the liquid surface tension. The transition to film formation and therefore wetting occurs for capillary numbers, which depend on the contact angle; the smaller the contact angle, the smaller the critical capillary number for the wetting transition. Since the viscosity of the polymer solutions used here depends on the shear rate, the capillary number will have to be redefined as $Ca = \eta(\dot{\gamma}) V / \sigma$, where $\eta(\dot{\gamma})$ is the viscosity at the specified shear rate $\dot{\gamma}$ (Supplementary Fig. 2). Here, the shear rate is fixed by the velocity and the thickness of the fluid underneath the flexible blade (Supplementary Fig. 4 and Supplementary Note 1).

Consider first the Newtonian fluid. At low velocities, the contact line is smooth with some undulations present. In fact the fluid dewets and this dewetting induces these sinusoidal-like deformations[13]. The length scale separating these undulations decreases with the driving velocity. Above a certain velocity or capillary number, these undulations become cusp-like, giving rise to rivulet formation, as seen before at the rear of drops flowing down a substrate[15]. These rivulets are unstable and small drops are produced, as seen in Fig. 2a. At even higher velocities, the cusp angle at the base increases and the increase in velocity, and therefore capillary number, gives rise to film formation and therefore wetting, as shown in Fig. 2a (Supplementary Figs 5 and 6 and Supplementary Note 1). This scenario has been observed before in dip-coating experiments[14]. In the wetting diagram presented in Fig. 2, the capillary number increases linearly with the plate velocity for the Newtonian fluid and two thresholds can be identified: the first threshold for cusp formation and the second threshold for film formation and wetting[14,15].

Consider now the polymer solution (PAM, see Methods). Because the shear viscosity decreases with shear rate as $\eta \sim \dot{\gamma}^{-0.8}$ (Supplementary Fig. 2), the capillary number varies

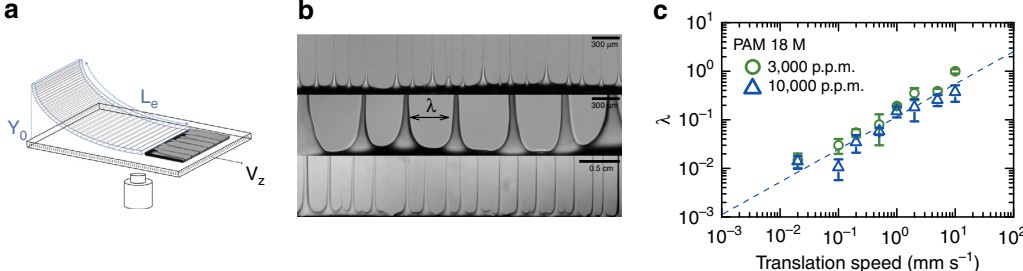

**Figure 1 | Deposition of a polymer solution.** (**a**) Schematic of the flexible blade set-up. A glass surface is translated below the blade, which is fixed to entrain fluid underneath the blade. The meniscus between the fluid and the surface resists wetting the latter and recedes as it is dragged by the bottom surface. (**b**) Photographs of filaments near the blade for different velocities ($V = 0.2, 0.5, 10 \text{ mm s}^{-1}$). Note the change in scale as the velocity is increased. (**c**) Wavelength versus deposition velocity for two different concentrations (3,000 and 10,000 p.p.m.) of PAM of molecular weight 18 M. The error bars in **c** represent the s.d. More details are shown in Supplementary Fig. 1.

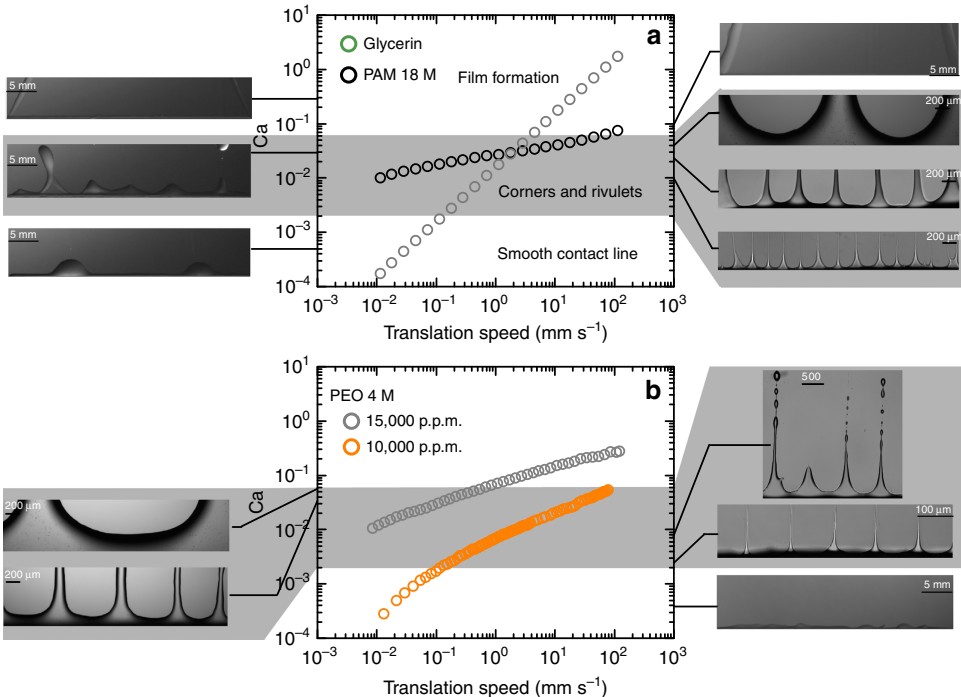

**Figure 2 | Capillary number versus velocity.** (**a**) For Glycerine and for a PAM solution at 10,000 p.p.m. For Glycerine the capillary number Ca is linear versus velocity. It is not for PAM because of shear thinning. The grey area designates the region where cusps form as the meniscus is being dragged by the bottom plate. At higher velocities and therefore higher values of Ca, a film forms on the surface. For PAM, the values of Ca reside mainly in the grey region, giving rise to cusps and filaments. Left images are for Glycerine, right images are for PAM. Panel (**b**) shows a similar plot for another polymer solution PEO of molecular weight 4 M at two different concentrations. By increasing the concentration the Ca can be made to reside mainly in the grey region and cusps giving rise to filaments can be made stable. For the lower concentration (right images), the filaments break up into drops as the characteristic time for filament break-up is smaller than for the higher concentration (left images).

much more slowly with the driving velocity. This slow variation makes for capillary numbers, which are basically between the two threshold values observed for the Newtonian fluid. Cusp formation therefore occurs even at low velocities and continues to exist at much higher ones. These cusps therefore occur over a much wider velocity range than for Glycerine. Film formation and therefore wetting for the polymer solution is observed only at the high-velocity end. Rivulets form for this solution, but they are not prone to drop formation, as shown in Fig. 2a, and remain as filaments. Polymer solutions are known to form filamentary structures stabilized by the extensional properties of the fluid[20]. Just like in droplet pinch-off experiments, the presence of polymers inhibits the break-up of the fluid necks and threads and gives rise to long slender filaments with a long lifetime[20,21] (Supplementary Fig. 7 and Supplementary Note 2). This property plays a similar role here and inhibits droplet formation from rivulets, thereby stabilizing the filaments emanating from the cusps. We believe that this filament stability is responsible for the long filaments observed here as the filaments persist only when the filament lifetime, measured in complementary experiments (Supplementary Fig. 7 and Supplementary Note 2), is long enough.

**Filament stability**. Two factors therefore conspire to give rise to robust filaments over an extended velocity range: shear thinning, which limits the range of capillary numbers to the cusp formation region, and the extensional properties of the polymer solution, which stabilize the filaments. To test this scenario, other polymer solutions can be prepared and made to undergo a similar transition. Consider the case of another polymer solution

(polyethylene oxide (PEO 10,000 p.p.m.), see Methods). The shear viscosity of this solution varies as $\eta \sim \dot\gamma^{-0.5}$ (Supplementary Fig. 2), so the capillary number varies more rapidly than for the first polymer solution, but more slowly than for the Glycerine solution, as seen in Fig. 2b. Now, for low velocities and therefore capillary numbers below the first threshold, only mild undulations are observed, as for Glycerine. For intermediate velocities for which the capillary number is greater than the first threshold, cusps can be seen and rivulet formation occurs. These rivulets are more unstable towards drop formation even though filament-like structures are still observed near the meniscus (Fig. 2b). At even higher velocities, the capillary number exceeds the second threshold and film formation occurs. The main difference between the two polymer solutions is the stability of filament structures; for the first solution the characteristic time for filament thinning as measured in complementary drop detachment experiments is larger than for the second solution (see complementary experiments in Supplementary Note 2 and Supplementary Fig. 7). If now we use a higher concentration of polymer (PEO 15,000 p.p.m.), the shear thinning behaviour turns out to be $\eta \sim \dot\gamma^{-0.66}$ (Supplementary Fig. 2) and the characteristic time for filament thinning in drop detachment experiments increases (see complementary experiments in Supplementary Note 2 and Supplementary Fig. 7). As seen in Fig. 2b, filament formation is again obtained (drop formation is inhibited) and the wetting phase diagram becomes close to that of the first solution. Increasing the filament lifetime increases the stability of the filaments in this case. The wavelength here again increases with the capillary number. The combination of shear thinning (to control the capillary number and give it values in the cusp formation region) and the capability of the solution to give rise to

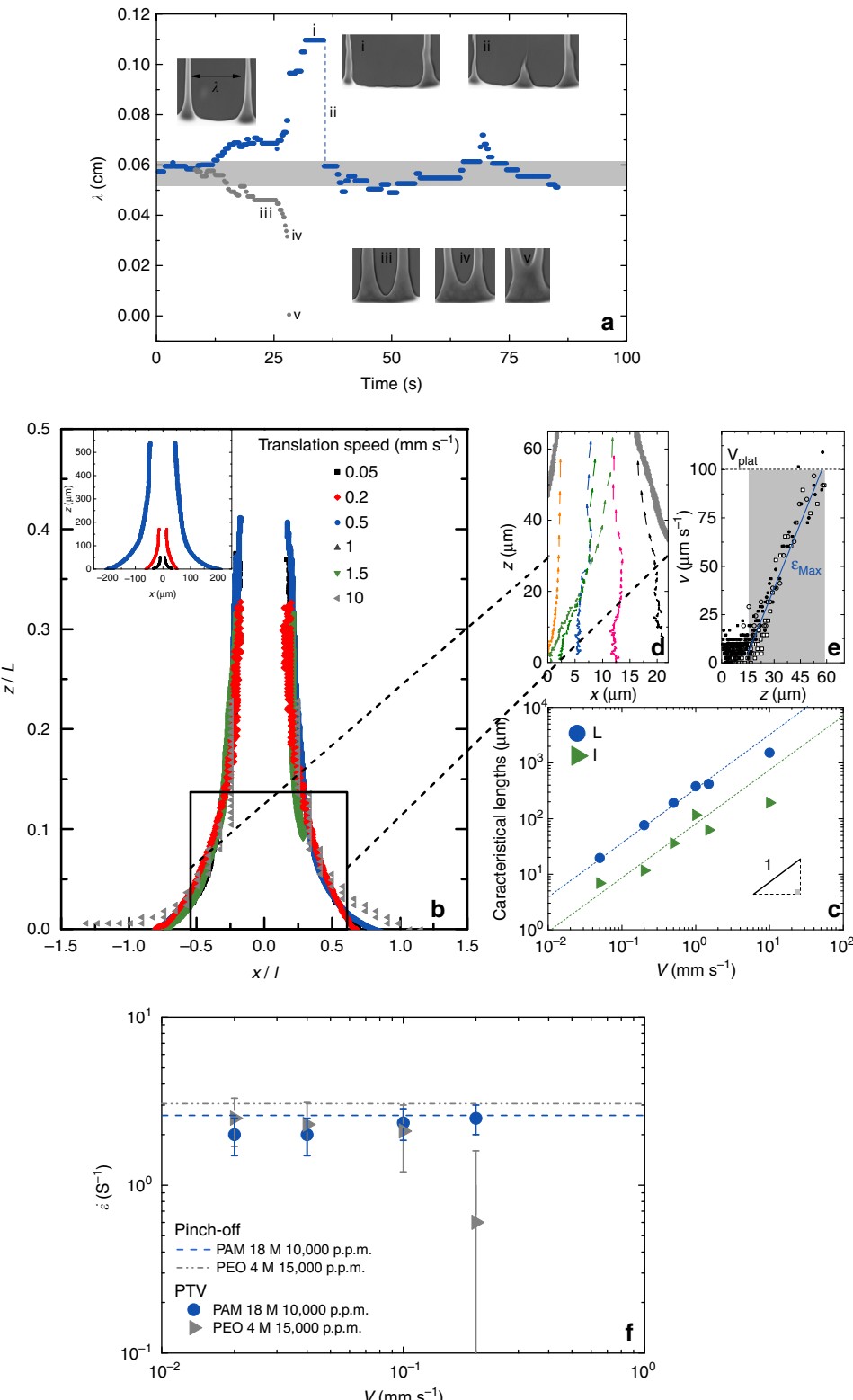

**Figure 3 | Filament characteristics.** (**a**) A schematic showing the distance between two filaments for $V = 0.25\,\mathrm{mm\,s^{-1}}$. As this distance increases with time, a new filament emerges between the two. In another event depicted in this graph, as two filaments approach each other, they merge and give rise to a single filament. The images show the state of the filaments at the different instants denoted by a small numbering. (**b**) The shape of the base of the filaments is portrayed for different velocities. This shape, shown in the inset, can be made dimensionless (main figure) by rescaling the vertical and horizontal axes using characteristic scales $L$ and $l$, respectively. (**c**) Both length scales vary linearly with velocity indicating a proportionality between the two lengths. The ratio $V/L$ then defines a time scale found in good agreement with the stretching rate in the filament itself and measured using particle tracking shown in (**d**,**e**) where the velocity of tracer particles in the direction of the filaments $v_z$ is plotted versus distance along the filament $z$. (**f**) The stretching rate $\dot{\varepsilon}$ obtained from $dv_z/dz$ turns out to be constant versus plate velocity $V$ and given by the characteristic time of the polymer solution measured using droplet pinch off experiments (see Supplementary Note 2 and Supplementary figures 6 & 7). The error bars in **f** represent the min and max values.

filament formation (through high extensional stresses[20,21]) is at the core of the patterns obtained. As we have shown above, tuning the fluid properties to have both characteristics brings forth the regular patterns in the case of both polymer solutions.

## Discussion

These filaments then form an organized pattern on the surface with a well-defined wavelength. While cusp formation and eventual formation of rivulets and droplets has been observed before, the spatial organization into an organized pattern has not been seen before. Further, the observed wavelength increases as the velocity of the receding contact line increases in stark contrast to the known fingering instabilities observed for advancing contact lines, where the wavelength is a decreasing function of velocity.

While no theory exists to describe the wavelength variation with velocity in our experiments, we believe that the dynamics of coalescence between neighbouring cusps and filaments attached as well as the viscoelastic character of the solutions used here is at the origin of these properties. Figure 3a illustrates this dynamics: as two filaments drift apart from each other, a third filament appears in the spacing between them. Further, and as two filaments approach each other, coalescence occurs. The creation of new filaments and the coalescence of nearby ones regulates the selection of the wavelength. This dynamics seems to be related to the shape of these filaments near their base; as two filaments approach each other, the lateral curvature of the contact line between the two filaments increases and a merger occurs (see Supplementary Movies SV01, SV02 and SV03 corresponding to the images of Fig. 1b). Figure 3b shows a close-up of this shape for different velocities. Note that the functional shape is self-similar and independent of velocity: rescaling the shape by two characteristic scales, $l$ for the width and $L$ for the length, as shown in Fig. 3b,c, collapses all filaments onto a universal shape. Deviations from this shape are observed near the base mostly due to the difficulty of detecting the meniscus and its fluctuations.

Another feature characterizes these filaments. Through measurements of the velocity of the fluid within these filaments and along the centreline, $v_z$ (Fig. 3d,e), the mean extension rate $dv_z/dz$ in the flow direction $z$ is independent of velocity $V$. This extension rate depends only on the characteristic time of these viscoelastic solutions, which we estimate from droplet detachment experiments as seen in Fig. 3f (refs 20–22) (Supplementary Figs 7 and 8 and Supplementary Note 2). This observation shows that filament dynamics and stretching, whether in suspended filaments as in droplet detachment where the solution forms long slender filaments that thin exponentially in time (Supplementary Fig. 7) or in slender filaments deposited on a surface, are similar and are controlled by the relaxation time of the polymer solution. By using the characteristic length of the filament $L$ and the velocity of the plate $V$, this mean extension rate can be estimated as $V/L$, indicating that $L$ should be proportional to the velocity $V$ to keep the extension rate constant. This is borne out experimentally in Fig. 3c. We believe that these two ingredients, the self-similar shape of the filaments and the fixed nature of the characteristic extension rate, coupled to the fact that if the spacing between two filaments is wide enough a new filament ensues, are crucial in determining the spatial organization of these patterns. Since the width of the fingers $l$ increases with their characteristic length $L$ and both increase with velocity (Fig. 3f) while keeping the extension rate constant, the wavelength will have to increase so that coalescence between fingers is avoided.

The above results and discussion show that by tuning the liquid properties, the deposition of the fluid on the surface in dip coating experiments or in blade-coating studies can be modified controllably. It is by taming the instability of receding contact lines towards cusp and droplet formation through the use of viscoelastic fluids that such control can be achieved. This tunability and controllability may open new possibilities and directions in coating flows. The spatially organized and tunable properties of the patterns found here can actually be useful as they may allow to print a variety of linear patterns. This capability to print such tunable and anisotropic structures may be used for printing optical gratings or making directional conductive coatings. To illustrate the possibilities offered by these results, we show the possibility to control the linear deposition of different additives (Supplementary Note 3). In Fig. 4a,b, we show

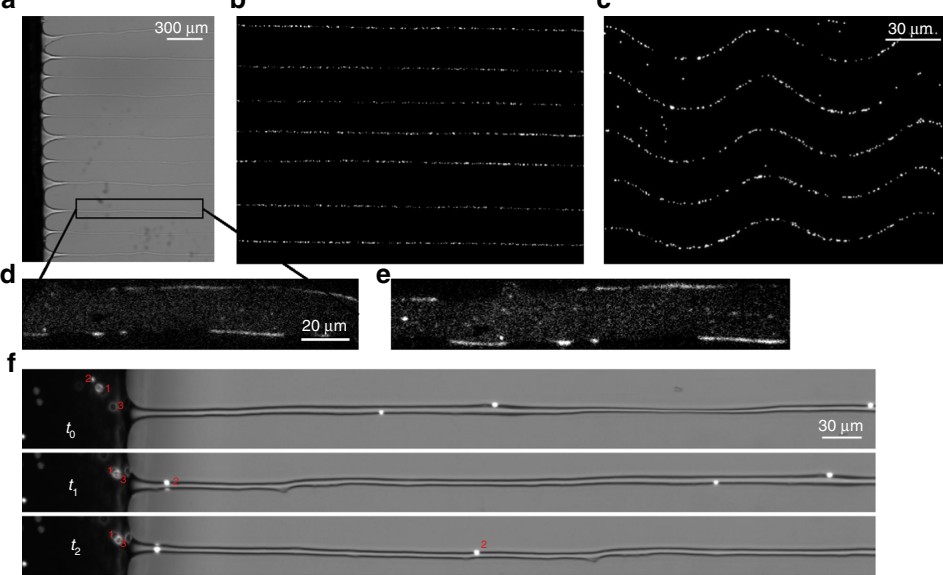

**Figure 4 | Useful patterns.** Photographs of the filaments (**a**) with embedded fluorescent colloidal particles of 1 μm in diameter (**b**). (**c**) Wavy filaments with the same fluorescent particles as in **b** obtained by translating the plate with a sinusoidal velocity in the plane of the substrate. (**d**,**e**) Stretched fluorescent DNA molecules, up to 1/3 of their total extension, embedded in the filaments of photo (**a**). (**f**) Filtering by size of 1 and 6 μm particles. The fluorescent 1 μm particles (such as particle 2) enter the thin filaments while the 6 μm (particles 1 and 3) do not. The three photographs at are times $t_0$, $t_1 = 10.9$ s, $t_2 = 21.3$ s.

that colloidal particles embedded into the polymer solution can be deposited on the substrate to form a pattern of oriented lines with regular spacings. Further and by simply using a sinusoidal velocity for the substrate, wavy patterns can be deposited, as seen in Fig. 4c. Along these lines, additional macromolecules such as DNA embedded into the polymer solution can also be deposited in a stretched configuration imposed by the direction of the filaments, as seen in Fig. 4d,e, suggesting the possibility of printing filaments with embedded anisotropy[23–29]. The tunability of the obtained structures both in wavelength and in dimensions also allows to use these patterns to filter different size additives. Here, particles of 6 and 1 μm diameter are mixed together in solution. As they approach the contact line, only the smaller particles manage to enter the thin filaments while the larger ones stay blocked near the meniscus, as has been anticipated theoretically[30]: particle filtering by size during deposition can therefore be achieved, as seen in Fig. 4f (Supplementary Movie SV04). The sizes to be filtered can actually be tuned by simply using higher or lower velocities for which the finger width can be increased or decreased (Figs 3 and 4). In short, the ability to change fluid properties using viscoelastic solutions opens new directions in the control of instabilities in coating flows.

## Methods

**Coating set-up.** The experimental set-up consists of a thin flexible blade (mylar sheet of dimensions $4 \times 3\,cm^2$ and thickness $200\,μm$) held fixed vertically. The spacing $y_0$ between the glass plate and the upper end of the flexible blade can be adjusted with a translation stage with $10\,μm$ precision; the length $L_e$ of this scraper can be also adjusted. The polymer solution is deposited on a hydrophobic glass substrate with dimensions $6 \times 15\,cm^2$ and $2\,mm$ thickness. The substrate is then translated in the $z$ direction with a precision motorized translation stage (Marzhauser Wetzlar SCAN IM $130 \times 100$) capable of moving at stable speeds ranging from 0.1 to $10\,mm\,s^{-1}$. This set-up is mounted on an inverted microscope stage (Axio observer A1 from Zeiss).

**Substrate preparation.** The glass substrates were coated with octadecyltri-chlorosilane to render them hydrophobic by following the protocol below: The glass plates were cleaned with an industrial cleaning agent (Decon at 2%) and rinsed with pure water. They were dried in an oven. To get rid of possible grease residues, we placed them in a plasma cleaner during 15 min. The substrates are placed during 30 min in a solution of iso-octane (2,2,4-trimethylpentane from Sigma Aldrich) with 1% of octadecyltrichlorosilane (from Sigma Aldrich). Otherwise, they are simply cleaned in a plasma cleaner to render them hydrophilic. The contact angles of our polymer solutions on the glass substrates were measured using photographs of small droplets (Supplementary Fig. 3). A summary of this data is given in Supplementary Table 1.

**Formulation of solutions.** Our experiments were performed with glycerine (99.9% from Sigma Aldrich), PAM and PEO solutions (from Polysciences and Sigma-Aldrich) dissolved in pure water. For PAM, 10 mM NaCl was added to the solution. For PAM solutions, we used polymers with two different molecular weights $M_W = 10 \times 10^6\,g\,mol^{-1}$ and $18 \times 10^6\,g\,mol^{-1}$. For PEO solutions, the molecular weight was $M_W = 4 \times 10^6\,g\,mol^{-1}$. Both polymer solutions were made at different concentrations $C_P$ from 3,000 to 15,000 parts per million by weight (p.p.m.). To carry out PTV measurements within the deposited structures, we seeded the solutions with small particles of 1 or 6 μm diameter (PMMA or polystyrene fluorescent particles from Polysciences and Molecular Probes) and tracked their trajectories from video imaging. The DNA T4 GT7 (from Nippon Gene) are made fluorescent thanks to the intercalator POPO3 (from Molecular Probes). It was added in the proportion of one to five base pairs. The DNA and polymer solutions are made with Tris EDTA buffer solution (from Fluka) with pH 8. The total length of DNA T4 with fluorescent molecules is estimated as 72 μm.

**Rheometry.** The rheological properties of the polymer solutions used have been measured with an ARG2 rheometer using a cone-plate geometry with 50 mm diameter cone and a 1° angle. The high concentration polymer solutions used are non-Newtonian and have a shear thinning behaviour with a viscosity $η$ decreasing versus the shear rate $\dot{γ}$ as a power law $η \sim \dot{γ}^{-0.8}$ for PAM solutions at high concentrations (10,000 p.p.m.), as $η \sim \dot{γ}^{-0.66}$ for a concentration of 3,000 p.p.m. and PEO 4 M at 15,000 p.p.m., and as $η \sim \dot{γ}^{-0.5}$ for PEO 4 M at 10,000 p.p.m (Supplementary Fig. 2). The higher the molecular weight and the concentration, the stronger the shear thinning.

**Imaging techniques.** We recorded movies of the contact line region at two different scales: using a microscope equipped with a camera (Hamamatsu Orca 2.8 and 4.0) and a sensitive camera (Hamamatsu EM-CCD), or simply a CCD camera (Giuppy Pro from R&D Visions) equipped with a macrolens, depending on the size and the spacing between the fingers. PTV measurements were performed with a homemade routine using Matlab and ImageJ.

**Data availability.** All relevant data presented in the paper is available upon request.

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

## Acknowledgements

We would like to acknowledge the LabEx AMADEus, ANR-10-LABX-0042-AMADEUS, along with the program of Excellence Initiative IdEx Bordeaux, grant no. ANR-10-IDEX-0003-02, for financial support. Also, we acknowledge the help of the machine shop (E. Maillard, S. Bosio and L. Haelman) for the manufacturing of the scraper set-up. H. Kellay acknowledges support from IUF.

## Author contributions

A.D. and R.H. carried out the experiments with the help of H.K. A.C. and H.K. designed the experiment. All authors contributed to data analysis and interpretations. H.K. wrote the paper with contributions from all authors.

## Additional information

**Competing financial interests:** The authors declare no competing financial interests.

