## [Peer Review File · Nature Communications]

Reviewers' comments:

Reviewer #1 (Remarks to the Author):

The authors report dip coating experiments that demonstrate a controlled pattern formation as narrow filaments form over an extended velocity range, and by contrasting Newtonian and viscoelastic fluids they are able to learn about important features of the viscoelastic material responsible for the observations. In particular, for viscoelastic fluids the authors show that shear thinning limits the range of capillary numbers to the cusp formation region and the extensional properties of the polymer solution stabilize the filaments against drop formation. The observed wavelength of the filaments increases as the velocity of the receding contact line (plate speed) increases. The authors have done a very good job characterizing the experiments, e.g. they find that there is a self-similar shape of the filaments and they argue that there is an approximately constant extension rate in the thin film. The authors are also able to provide some potential applications by controlling the filament spacing with speed and entraining materials (particles, DNA) in the filaments. I have not seen results quite like this before so I think the authors are pointing out new important directions for coating flows and highlighting how properties of the fluid may be selected to give certain pattern-forming characteristics. I think readers will find the work to be interesting and I recommend publication.

Additional remarks:

- 1) Label λ on figure 1b
- 2) Figure 1c: identify the blue versus the green symbols.
- 3) Figure 2 caption: "The lower concentration (right images) although gives rise to filament formation" - needs re-wording
- 4) p. 2: "which depend on the contact angle, the smaller the contact angle, the smaller the critical capillary number for the wetting transition." -> the comma should be replaced by a ;
- 5) p. 3: "Film formation for the polymer solution occurs only at the high velocity end." It is not clear from the wording whether or not the authors have actually observed a continuous film for the viscoelastic solutions. I think the answer is yes since this seems to be shown in figure 2 and discussed on page 3 but some of the wording could be clarified.
- 6) p. 3: "Consider the case of another polymer solution" - at least say clearly in the main text what this solution is.
- 7) Some of the panels in figure 3 are too small (e.g. figure 3a). Many of the figure labels were too small to read easily.
- 8) Figure 4 shows a type of filtering of particles by size. The experiments remind me of a 2014 PRL showing simulations of how particles can be entrained, or not, into thin films whose thickness changes with pulling speed of the plate (Colosqui et al. Phys. Rev. Lett. 110, 188302).

Reviewer #2 (Remarks to the Author):

The major claim of this paper is that by using a non-Newtonian fluid one could produce a coating

pattern composed of thin liquid threads with controlled separation (versus a thin film or multiple droplets), which appears to be novel and reasonable well justified. While the results are interesting, they are certainly not groundbreaking. Hence the significance of the paper is mostly in the potential applications of the technique (several of which have been identified by the authors) rather than the fundamental importance of the phenomena studied. As a result, this paper would be of interest to researchers in the fields of non-Newtonian fluids, microfluidics, and free-surface flows, but likely not to a broad spectrum of readers even in the fluid dynamics community.

Here are some specific issues that the authors should address:

1. "...the spatial organization into an organized pattern has not been anticipated." It seems to be a very natural result. Why does it appear unanticipated to the authors?
2. "The stretching rate obtained from dv_z/dz turns out to be constant versus plate velocity V and given by the characteristic time of the polymer solution measured using droplet pinch off experiments." This sentence is not particularly clear. This relation is quite important and should be properly discuss in the main text rather than in a figure caption most of the readers are likely skip completely. I did not notice it until after reading the supplementary material. It's worth explaining why the rates are expected to be the same. This is FAR from obvious.
3. "We believe that these two ingredients, the self similar shape of the filaments and the fixed nature of the characteristic extension rate determine the spatial organization of these patterns." I don't see sufficient evidence supporting this claim. In fact, the universal scaling clearly breaks down at the base of the filaments, which is the region of the film affected by the contact line instability responsible for the filament spacing. The linear scaling of filament width with velocity limits the smallest spacing achievable, but the actual spacing could be much larger.
4. "...print a variety of linear or wavy patterns..." The reference to wavy patterns is sudden, unexpected, not properly explained, and, on closer inspection, rather trivial (it is due to transverse motion of the plate). Ditto for the discussion of suspended particles. In fact, the entire last paragraph of the main section of the paper is extremely poorly integrated with the rest of the paper and lacks rigor and depth. The paper needs a proper conclusions section/paragraph instead.

Minor comments:

5. "Coating surfaces with fluids and complex fluids is essential..." What are the fluid in the first instance?
6. Fig. 1c caption: Explain color-coding
7. Most figures: scale bars' font is too small to be readable
8. Fig. 2 top left panel has contrast so low it is essentially unusable
9. "Consider the case of another polymer solution..." Identify the polymer in the text.
10. Fig. 3 panels are unreadable.
11. Why use bizarre notations such as " $\text{mm}\cdot\text{s}^{-1}$ " instead of the natural mm/s ?

REVIEWERS' COMMENTS:

Reviewer #2 (Remarks to the Author):

The authors have adequately addressed all of my comments and I can now recommend publication of the paper.

Point by point response to the referees comments: Reviewer #1 (Remarks to the Author):

The authors report dip coating experiments that demonstrate a controlled pattern formation as narrow filaments form over an extended velocity range, and by contrasting Newtonian and viscoelastic fluids they are able to learn about important features of the viscoelastic material responsible for the observations. In particular, for viscoelastic fluids the authors show that shear thinning limits the range of capillary numbers to the cusp formation region and the extensional properties of the polymer solution stabilize the filaments against drop formation. The observed wavelength of the filaments increases as the velocity of the receding contact line (plate speed) increases. The authors have done a very good job characterizing the experiments, e.g. they find that there is self similar shape of the filaments and they argue that there is an approximately constant extension rate in the thin film. The authors are also able to provide some potential applications by controlling the filament spacing with speed and entraining materials (particles, DNA) in the filaments. I have not seen results quite like this before so I think the authors are pointing out new important directions for coating flows and highlighting how properties of the fluid may be selected to give certain pattern forming characteristics. I think readers will find the work to be interesting and I recommend publication.

Additional remarks:

- 1) Label λ on figure 1b
- 2) Figure 1c: identify the blue versus the green symbols.
- 3) Figure 2 caption: "The lower concentration (right images) although gives rise to filament formation" - needs re-wording
- 4) p. 2: "which depend on the contact angle, the smaller the contact angle, the smaller the critical capillary number for the wetting transition." -> the comma should be replaced by a ;
- 5) p. 3: "Film formation for the polymer solution occurs only at the high velocity end." It is not clear from the wording whether or not the authors have actually observed a continuous film for the viscoelastic solutions. I think the answer is yes since this seems to be shown in figure 2 and discussed on page 3 but some of the wording could be clarified.
- 6) p. 3: "Consider the case of another polymer solution" - at least say clearly in the main text what this solution is.
- 7) Some of the panels in figure 3 are too small (e.g. figure 3a). Many of the figure labels were too small to read easily.
- 8) Figure 4 shows a type of filtering of particles by size. The experiments remind me of a 2014 PRL showing simulations of how particles can be entrained, or not, into thin films whose thickness changes with pulling speed of the plate (Colosqui et al. Phys. Rev. Lett. 110, 188302).

Response to referee 1:

We thank the referee for finding our work novel and of interest and for recommending publication in Nature Communications. We also thank the referee for all the suggestions for improving the paper and the figures. The revised version of the paper takes into account every item listed by the referee as well as the inclusion of the reference to theoretical work on how particles interact with a meniscus.

Reviewer #2 (Remarks to the Author):

The major claim of this paper is that by using a non-Newtonian fluid one could produce a coating pattern composed of thin liquid threads with controlled separation (versus a thin film or multiple droplets), which appears to be novel and reasonable well justified. While the results are interesting, they are certainly not groundbreaking. Hence the significance of the paper is mostly in the potential applications of the technique (several of which have been identified by the authors) rather than the fundamental importance of the phenomena studied. As a result, this paper would be of interest to researchers in the fields of non-Newtonian fluids, microfluidics, and free-surface flows, but likely not to a broad spectrum of readers even in the fluid dynamics community.

Here are some specific issues that the authors should address:

1. "...the spatial organization into an organized pattern has not been anticipated." It seems to be a very natural result. Why does it appear unanticipated to the authors?
2. "The stretching rate obtained from dv_z/dz turns out to be constant versus plate velocity V and given by the characteristic time of the polymer solution measured using droplet pinch off experiments." This sentence is not particularly clear. This relation is quite important and should be properly discuss in the main text rather than in a figure caption most of the readers are likely skip completely. I did not notice it until after reading the supplementary material. It's worth explaining why the rates are expected to be the same. This is FAR from obvious.
3. "We believe that these two ingredients, the self similar shape of the filaments and the fixed nature of the characteristic extension rate determine the spatial organization of these patterns." I don't see sufficient evidence supporting this claim. In fact, the universal scaling clearly breaks down at the base of the filaments, which is the region of the film affected by the contact line instability responsible for the filament spacing. The linear scaling of filament width with velocity limits the smallest spacing achievable, but the actual spacing could be much larger.
4. "...print a variety of linear or wavy patterns..." The reference to wavy patterns is sudden, unexpected, not properly explained, and, on closer inspection, rather trivial (it is due to transverse motion of the plate). Ditto for the discussion of suspended particles. In fact, the entire last paragraph of the main section of the paper is extremely poorly integrated with the rest of the paper and lacks rigor and depth. The paper needs a proper conclusions section/paragraph instead.

Minor comments:

5. "Coating surfaces with fluids and complex fluids is essential..." What are the fluid in the first instance?
6. Fig. 1c caption: Explain color-coding
7. Most figures: scale bars' font is too small to be readable
8. Fig. 2 top left panel has contrast so low it is essentially unusable
9. "Consider the case of another polymer solution..." Identify the polymer in the text.

10. Fig. 3 panels are unreadable.

11. Why use bizarre notations such as "mm.s-1" instead of the natural mm/s?

Response to referee 2.

We thank the referee for his/her comments. The referee acknowledges that the results are novel, interesting, and likely to interest different communities. We regret however that the referee does not acknowledge that the significance of the results does not go beyond the potential for applications. Our results and diagnostics bring forth several novel features for how the meniscus of a viscoelastic fluid adapts its shape to the driving imposed by a moving solid surface. These features have neither been identified before nor have they been observed. Besides the novelty of the observed patterns, we have pointed out the essential aspects such as the self similar shapes near the meniscus and the constant extension rate within the filaments as well as the dynamics of coalescence between fingers. We believe that these elements can bring further insight into any attempt to model such a complex and fundamental problem where viscoelasticity and contact line dynamics are present. An important fundamental aspect of our work is to point to the universal character of the forced wetting diagram established in our Figure 2 as it can be described following the framework for Newtonian fluids albeit for the capillary number which takes the rheology into account. The observed patterns are then recast into this framework with the elastic effects entering into play to favor filaments by inhibiting droplet formation. As far as we know, such a diagram has not been established before. We here answer each specific comment raised by the referee and the text has been modified to take these comments into account.

Responses to specific comments:

1. We have said that the organization into well defined filaments with a well defined wavelength has not been anticipated before. We now say that it has not been observed before. We still think that obtaining such a well organized pattern has not been discussed explicitly before and did not find specific mention of this. It is only by inhibiting droplet formation, by stabilizing the filament, and reducing the variation of the capillary number that such an observation has been made possible. As far as we know, there are no works mentioning such a possibility for Newtonian fluids nor for viscoelastic fluids.

2. We now clarify this sentence. We agree with the referee that this relation is not obvious. The same problem arises in droplet detachment experiments using viscoelastic polymer solutions. The detachment gives rise to long slender filaments for which the stretching rate is constant and given by the relaxation time of the polymer solution. This has been dealt with on theoretical grounds (see McKinley et al. and Hinch et al.). We have actually used this to obtain the relaxation times of the solutions. There is no expectation for a filament like the ones we observe but nonetheless our observations suggest that they too seem to select a stretching rate which is given by the solution relaxation time. We explain this further in the revised version.

3. The referee is right, these two ingredients determine the smallest spacing, as for the large spacing, it is the nucleation of new filaments which is at work. We now specify that this nucleation of new filaments is essential. We had already said this but in a separate paragraph where we describe Fig. 3a and where we say that when two filaments drift apart from each other, a new filament emerges in the spacing between them. The scaling we show in Fig. 3 becomes poor near the base for a few examples but this is due mostly to the difficulty of detecting the contour of the meniscus in this region which is prone to fluctuations. We have added this to the text in the revised version.

4. We have modified this last paragraph so that it fits in better with the rest of the paper. The rationale behind this paragraph is to point out that such patterns can actually be useful for printing anisotropic patterns using molecules or using colloids. Further, and because of the tunability of the obtained patterns one can use them for particle filtering. This paragraph therefore brings forth the motivation announced from the start, namely coating and printing using dip or blade coating techniques.

Minor points:

5. We now specify Newtonian fluids
6. We have added the color coding to this figure. We apologize for this omission.
7. We have resized all scale bars.
8. Left panel has been made slightly more contrasted.
9. We now specify the polymer used.
10. We have split Fig 3 into separate panels for better readability.
11. We have corrected this mistake in units (s^{-1} is meant to be s^{-1} and the units have been modified to mm/s).

Again, thank you for your careful reading of our manuscript and for raising points which helped us make the paper clearer. We hope these changes and our answers convince you that our paper is suitable for Nat. Comm.